# *Salvia plebeia* R.Br. and Rosmarinic Acid Attenuate Dexamethasone-Induced Muscle Atrophy in C2C12 Myotubes

**DOI:** 10.3390/ijms24031876

**Published:** 2023-01-18

**Authors:** Jae-Yong Kim, Hye Mi Kim, Ji Hoon Kim, Shuo Guo, Do Hyun Lee, Gyu Min Lim, Wondong Kim, Chul Young Kim

**Affiliations:** College of Pharmacy and Institute of Pharmaceutical Science and Technology, Hanyang University, Ansan 426-791, Gyeonggi-do, Republic of Korea

**Keywords:** *Salvia plebeia* R.Br., Rosmarinic acid, dexamethasone, atrophy, C2C12

## Abstract

Skeletal muscle atrophy occurs when protein degradation exceeds protein synthesis and is associated with increased circulating glucocorticoid levels. *Salvia plebeia* R.Br. (SPR) has been used as herbal remedy for a variety of inflammatory diseases and has various biological actions such as antioxidant and anti-inflammatory activities. However, there are no reports on the effects of SPR and its bioactive components on muscle atrophy. Herein, we investigated the anti-atrophic effect of SPR and rosmarinic acid (RosA), a major compound of SPR, on dexamethasone (DEX)-induced skeletal muscle atrophy in C2C12 myotubes. Myotubes were treated with 10 μM DEX in the presence or absence of SPR or RosA at different concentrations for 24 h and subjected to immunocytochemistry, western blot, and measurements of ROS and ATP levels. SPR and RosA increased viability and inhibited protein degradation in DEX-treated C2C12 myotubes. In addition, RosA promoted the Akt/p70S6K/mTOR pathway and reduced ROS production, and apoptosis. Furthermore, the treatment of RosA significantly recovered SOD activity, autophagy activity, mitochondrial contents, and APT levels in DEX-treated myotubes. These findings suggest that SPR and RosA may provide protective effects against DEX-induced muscle atrophy and have promising potential as a nutraceutical remedy for the treatment of muscle weakness and atrophy.

## 1. Introduction

Skeletal muscle is the most abundant and essential tissue in the human body and involves mobility, protection, physical activities, and glucose with the lipid metabolism [1]. Furthermore, it has significant functions, including postural and balance maintenance, respiratory mechanics, metabolism, whole-body homeostasis, insulin-stimulated glucose uptake and storage, and repair after injury [2]. In general, healthy skeletal muscle has a good balance between protein synthesis and protein degradation, which plays an important role in maintaining muscle mass. 

Muscle atrophy is the loss of skeletal muscle mass as a result of increased myofibrillar proteolysis and decreased synthesis [3]. It is caused by a variety of factors, including aging, disuse, various chronic disease, and medications such as glucocorticoids [4]. Muscle atrophy reduces the patient’s movement independence and quality of life, and increases morbidity and mortality [5,6]. Recently, muscle atrophy has not only imposed an additional financial burden on the healthcare system but also has become a global social problem as the human lifespan has been extended with advances in medicine [7]. Therefore, it is important to maintain healthy muscles, and the prevention or cure of muscle atrophy is necessary for preventing various diseases and achieving healthy aging and life.

Numerous pathological conditions characterized by skeletal muscle atrophy are associated with an increase in circulating glucocorticoid levels [8]. Glucocorticoids cause an imbalance between the rate of protein synthesis and protein degradation. Dexamethasone (DEX), a synthetic glucocorticoid, is widely used to mediate proteolytic muscle atrophy in both in vivo and in vitro models [9]. It has been reported that DEX promotes skeletal muscle atrophy by inducing protein degradation through upregulation of the muscle-specific ubiquitin E3-ligases atrophy gene-1/muscle atrophy F-box (Atrogin-1/MAFbx) and muscle ring-finger protein 1 (MuRF1) and decreasing muscle protein synthesis through decreased phosphorylation of muscle synthesis factors including Akt, mammalian target of rapamycin (mTOR), and ribosomal protein S6 kinase 1 (S6K1) [10,11]. Although DEX is widely and effectively used as a therapeutic agent due to its powerful anti-inflammatory and anti-shock properties, as well as its protection against autoimmune diseases, high doses or long-term consumption of DEX induces severe muscle atrophy [12]. However, regardless of the clinical consequences of muscle atrophy, there is currently no drug treatment available. Therefore, it is required to discover substances and explore strategies to prevent DEX-induced muscle protein degradation and increase muscle protein synthesis.

Salvia plebeia R. Br. (SPR), belonging to the labiaceae family, is widely distributed in various counties such as Korea, India, China, Japan, and Australia [13]. SPR has been used in Korea as a traditional medicine to treat inflammation, hepatitis, rheumatoid arthritis, common cold, cough, flu, and asthma [14]. Furthermore, previous studies have reported that SPR presents a multimode of biological activities such as antioxidant, antimicrobial, anti-inflammatory, antiviral, and antitumor [15,16]. SPR has been known to contain diverse constituents, including flavonoids, monoterpenoids, sesquiterpenoids, diterpenoids, triterpenes, and phenolic acids [14]. Rosmarinic acid (RosA) is a major phenolic acid from SPR, which has various biological activities including antiviral, antibacterial, antioxidant, antimutagenic, and anti-inflammatory activities [17]. Although SPR and RosA have various biological roles in the human body, their anti-muscle atrophic function has not yet been elucidated. Therefore, we hypothesized that SPR and RosA would prevent DEX-induced muscle atrophy in C2C12 myotubes and investigated, for the first time, the protective effects of SPR and RosA against DEX-mediated muscle atrophy in C2C12 myotubes and its molecular mechanisms.

## 2. Results

### 2.1. Effects of SPR on Viability and Atrophy in DEX-Induced C2C12 Myotubes 

To compare the protective effect of SPR against DEX-induced cytotoxicity, fully differentiated C2C12 myotubes were treated with 10 µM DEX in the presence or absence of various concentrations of SPR (1, 5, and 10 µg/mL) for 24 h. DEX-treated myotubes significantly reduced cell viability compared to the control (Figure 1A), whereas co-treatment with 10 µg/mL of SPR with DEX significantly increased the viability compared with DEX-treated C2C12 myotubes (Figure 1A). Myosin heavy chain (MHC) immunostaining was performed to investigate a reliable phenotype of C2C12 myotubes (Figure 1B). Treatment of DEX remarkably reduced the MHC-positive area (Figure 1B,C), diameters (Figure 1B,D), and fusion index (Figure 1B,E) compared to the control. However, co-treatment of SPR (1, 5, and 10 µg/mL) with DEX significantly inhibited DEX-mediated reduction in the MHC-positive area, diameters, and fusion index capacity in C2C12 myotubes (Figure 1B–E), respectively. Especially, DEX-treated with SPR (1, 5, and 10 µg/mL) exhibited almost the same level of the MHC-positive area as the control group (Figure 1B,C). In addition, DEX significantly decreased MHC protein expression as well as increased atrogin-1 and MuRF1 protein expression (Figure 1F–I). On the other hand, MHC expression was significantly recovered with the addition of SPR at all concentrations (Figure 1F,G). Notably, atrogin-1 and MuRF1 levels were reduced similarly to the control by the SPR treatment (Figure 1F,H,I). These results suggest that SPR prevents DEX-induced cytotoxicity and muscle atrophy in C2C12 myotubes.

### 2.2. Effects of RosA on Viability and Atrophy in DEX-Treated C2C12 Myotubes 

We separated EtOH crude SPR extract into 11 fractions using centrifugal partition chromatography (CPC) (Appendix A), which is able to load a large volume of samples, high yields, and purity [18]. Next, the chromatograms of the 11 fractions were analyzed using the HPLC system, and fractions VII, VIII, and IX were found to be relatively major components (Appendix A). In addition, we evaluated that treatment with fraction VII significantly recovered the DEX-mediated decrease in cell diameters of C2C12 myotubes (Appendix A). Finally, we confirmed that fraction VII is rosmarinic acid through literature and comparative analysis with a standard reagent (Appendix A).

Co-treatment RosA (at 1 and 5 µM) with DEX significantly recovered the cell viability compared to the DEX-treated myotubes (Figure 2A). MHC immunostaining results showed that 1 µM and 5 µM RosA treatment markedly prevented the reduction of the MHC-positive area, diameter, and fusion index caused by DEX (Figure 2B–E). In particular, the diameter and fusion index at 1 µM RosA treated group showed almost the same level as the control (Figure 2D,E). In addition, co-treatment of RosA (1 and 5 µM) and DEX (10 µM) showed significantly increased MHC protein levels compared with the DEX alone group (Figure 2F,G). On the other hand, DEX-treated C2C12 myotubes significantly increased the expression of atrogin-1, MuRF1, and FoxO3a compared to the control group (Figure 2F,H–J). Compared to the DEX-treated group, treatment with RosA significantly reduced the expression of atrogin-1, MuRF1, and FoxO3a (Figure 2H–J). Surprisingly, the MHC protein level of RosA-treated myotubes was similar to or higher than that of the control group, and the expression level of atrogin-1 and MuRF1 protein was lower than that of the control group. These results suggest that RosA effectively prevents DEX-induced C2C12 myotube cytotoxicity and prevents muscle atrophy via increased related muscle differentiation protein and decreased muscle degradation proteins in C2C12 myotubes.

### 2.3. Effects of RosA on the Akt/mTOR/p70S6K Pathway in C2C12 Myotubes

To evaluate the effect of RosA on the muscle protein synthesis pathway against DEX treatment, the phosphorylation of Akt, mTOR, and p70S6K were measured in C2C12 myotubes. The phosphorylation of Akt, mTOR, and p70S6K of DEX-treated myotubes were significantly reduced compared with the control (Figure 3A–D). However, the phosphorylation of Akt, mTOR, and p70S6K significantly increased by treatment with 5 µM RosA in the presence of DEX (Figure 3B–D). Furthermore, the phosphorylation of Akt and p70S6K were remarkably increased at low concentrations of RosA (1 µM) treated in the presence of DEX in C2C12 myotubes (Figure 3B,D). Surprisingly, despite DEX treatment, the phosphorylation of Akt, mTOR, and p70S6K levels in the RosA-treated group was similar to or higher than in the control group.

### 2.4. Effects of RosA on *Reactive Oxygen Species* (ROS) Production, *Superoxide Dismutase* (SOD) Activity, and Apoptosis in DEX-Treated C2C12 Myotubes

CM-H_2_DCFDA staining showed that cellular ROS production was significantly increased in DEX-treated C2C12 myotubes compared with the control (Figure 4A,B). However, the production of ROS in DEX-treated myotubes was significantly reduced in the presence of RosA (Figure 4A,B). SOD is a crucial enzyme required to remove ROS. Whereas 10 µM DEX reduced the activity of SOD compared to the control, co-treatment with RosA and DEX significantly reversed the inhibition of SOD activities caused by DEX (Figure 4C). In addition, western blotting analysis was performed to demonstrate the effects of RosA on the expression of apoptosis-regulated proteins such as anti-apoptotic Bcl-2 protein and pro-apoptotic BAX protein. As shown in Figure 4D,E, the expression of Bcl-2 protein was decreased by DEX treatment, whereas the expression of BAX protein was increased. However, changes in Bcl-2 and BAX protein levels were restored by RosA treatment (Figure 4D,E). In particular, the expression of BAX was lower in the 1 µM and 5 µM RosA-treated groups than in the control group, and the expression of Bcl-2 was higher in the 5 µM RosA-treated group than in the control group, respectively (Figure 4D,E). Furthermore, Annexin V-FITC/Propidium iodide (PI) double fluorescent staining was performed to determine myotubes apoptosis (Figure 4F). As shown in Figure 4F,G, Annexin-V and PI-positive C2C12 myotubes were increased by DEX treatment. However, combined RosA and DEX treatment significantly reduced C2C12 myotubes stained with Annexin-V and PI compared to the DEX-treated group (Figure 4F,G). These results indicate that RosA significantly reduces ROS production, apoptosis, and necrosis and increases SOD activity in DEX-treated C2C12 myotubes.

### 2.5. Effects of RosA on Autophagy in DEX-Treated C2C12 Myotubes

To evaluate the role of RosA in autophagy regulation, the expression levels of autophagy-related proteins, including the microtubule-associated protein light chain 3 (LC3), Beclin1, and p62, were measured by western blotting. DEX-treated C2C12 myotubes significantly inhibited autophagy through decreased LC3-II/LC3-I ratio and Beclin1 expression and increased p62 expression compared to the control group (Figure 5A–D). However, treatment with RosA (1 and 5 μM) significantly restored autophagy activity by inhibiting DEX-induced autophagy-related protein changes in C2C12 myotubes (Figure 5A–D). These results indicate that RosA enhances autophagy activity in DEX-treated C2C12 myotubes.

### 2.6. Effects of RosA on Mitochondrial Content and ROS and ATP Level in DEX-Treated C2C12 Myotubes

To assess the effect of RosA on the quantitative and qualitative improvement of mitochondria in DEX-treated C2C12 myotubes, we measured mitochondrial content and ATP levels. DEX treatment markedly reduced mitochondrial content and ATP levels compared to the controls (Figure 6A–C). Compared with DEX-treated C2C12 myotubes, the treatment of 1 and 5 μM RosA significantly prevented the reduction of mitochondrial content (Figure 6A,B). In addition, the ATP level was significantly restored by the addition of 5 μM of RosA (Figure 6C). Furthermore, we evaluated mitochondrial ROS levels using MitoSOX staining in C2C12 myotubes. DEX treatment increased the production of mitochondrial ROS in myotubes (Figure 6D,E). However, co-treatment with RosA markedly inhibited the elevated production of mitochondrial ROS triggered by DEX (Figure 6D,E). These results suggest that RosA prevents DEX-induced muscle atrophy by the improvement of mitochondria, both quantitatively and qualitatively.

## 3. Discussion

From ancient times to the present, natural plants and their compounds have been studied as candidates for nutritional supplements and drugs for various diseases due to their advantages of low side effects, strong activity, and structural diversity [19]. *Salvia plebeia* R. Br (SPR). has various pharmacological activities and has been used as a traditional medicine to treat numerous diseases [14]. Rosmarinic acid (RosA), a phenolic acid, is the main constituent of SPR and has various biological activities [17]. However, despite these diverse activities, there has been no report on the effects and molecular mechanism of SPR and RosA on muscle atrophy. 

Here, we evaluated the preventive effects and action mechanism of SPR and RosA against DEX-induced muscle atrophy in C2C12 myotubes. Many studies have shown that DEX-treated C2C12 myotubes exhibit a significant decrease in cell viability, diameters, MHC-stained area, and fusion index [20,21]. These changes are representative features of the C2C12 myotube atrophy model induced by DEX. However, treatment with various natural products, such as myricanol, quercetin, and fucoxanthin, prevents these changes by DEX in C2C12 myotubes [20,22,23]. In the present study, SPR and RosA inhibited the cytotoxicity induced by 10 μM DEX (Figure 1A and Figure 2A), but SPR and RosA did not affect the hypertrophic effect of C2C12 myotubes (Appendix A). In addition, changes in C2C12 myotubes morphology by DEX were determined using MHC immunostaining, a representative marker of myotube differentiation. As shown in Figure 1B–E and Figure 2B–E, the MHC-stained area, diameters, and fusion index of C2C12 myotubes were reduced by DEX treatment but were significantly recovered by SPR or RosA treatment. These results suggest that SPR and RosA effectively prevent DEX-induced cytotoxicity and the inhibition of C2C12 myotube differentiation. 

It is well known that MHC is a key developmental regulator of skeletal muscle formation and differentiation, and that upregulation of MHC expression promotes skeletal muscle differentiation [24]. On the other hand, the muscle-specific E3 ubiquitin ligases muscle RING finger 1 (MuRF1) and muscle atrophy F-box (MAFbx)/atrogin-1 are major markers of muscle atrophy and play a crucial role in muscle protein degradation [10]. MuRF1 and atrogin-1 expression are regulated by FoxO3a, which translocates to the nucleus as a transcription factor upon dephosphorylation, eventually upregulating the expression of MuRF1 and atrogin-1 [25]. Previous studies have consistently reported that DEX treatment inhibits MHC protein expression and elevates the expression of MuRF1, Atrogin-1, and FoxO3a in C2C12 myotubes [25,26]. Our results showed that the treatment of DEX in C2C12 myotubes markedly reduced MHC expression and highly elevated protein levels of atrogin-1, MuRF1, and FoxO3a compared to the control (Figure 1F–I and Figure 2F–J). However, co-treatment of SPR and DEX significantly increased the expression of MHC and decreased the expression of atrogin-1, and MuRF1 (Figure 1F–I). In addition, compared with DEX-treated myotubes, RosA treatment significantly increased MHC expression and suppressed MuRF1, Atrogin-1, and FoxO3a, respectively (Figure 2F–J). In particular, even with DEX treatment, the C2C12 myotubes treated with RosA (at 1 and 5 μM) showed higher MHC expression than the control group, and the expression of atrogin-1 and MuRF1 were lower than that of the control group. These results suggest that RosA may prevent DEX-induced atrophy in C2C12 myotubes through an increase in muscle differentiation and inhibition of muscle degradation. 

The Akt/mTOR/p70S6K signaling pathway plays an important role in the process of cell proliferation and protein synthesis [27,28]. The serine/threonine kinase Akt and the mammalian target of rapamycin (mTOR) are important regulators of various cellular functions, including survival, growth, and differentiation [29,30]. In addition, a mitogen-activated Ser/Thr protein kinase, 70 kDa ribosomal S6 kinase (p70S6K), is an important factor involved in protein synthesis in mammalian cells [31]. Thus, upregulation of the AKT/mTOR/p70S6K signaling pathway is important for preventing muscle atrophy. In the present results, 10 µM DEX-treated C2C12 myotubes significantly decreased the phosphorylation of Akt, mTOR, and p70S6K compared to the control (Figure 3). However, the relative levels of phosphorylation of Akt and p70S6K were significantly increased by RosA treatment (at 1 and 5 µM) compared to the DEX-treated group (Figure 3A,B,D). The mTOR phosphorylation was also markedly improved by RosA (at 5 µM) compared to the DEX-treated group (Figure 3A,C). Especially, C2C12 myotubes treated with DEX and RosA (1 and 5 μM) had comparable or higher phosphorylation levels of Akt and p70S6K than controls. In addition, the phosphorylation of mTOR was similar to the control in C2C12 myotubes treated with 5 μM RosA. These results suggest that RosA upregulates the anabolism of muscle-specific proteins.

The production of ROS accelerates protein degradation in muscle fibers and inhibits protein synthesis during periods of skeletal muscle inactivity [32]. In addition, it has been reported that the overproduction of ROS is closely related to DEX or fasting-induced skeletal muscle atrophy [22,33]. Previous studies have reported that the ROS levels were elevated by DEX treatment in various cell lines such as C2C12 myotubes, osteoblast-like cells (MC3T3-E1), and human umbilical vein endothelial cells (HUVEC) [22,34,35]. According to recent studies, several natural products, such as *Valeriana fauriei* and morin, significantly inhibited ROS production in DEX-treated C2C12 myotubes [21,36]. Our results showed that the ROS productions in DEX-treated C2C12 myotubes were remarkably higher than that of the control (Figure 4A,B), which is consistent with the previously published results. On the other hand, the increase in ROS production by DEX was significantly decreased when treated with 1 μM and 5 μM RosA (Figure 4A,B). SOD is a key enzyme for scavenging ROS and reducing ROS accumulation [37]. Kim et al. reported that SOD activity was reduced in the muscle tissue of atrophy-induced mice by DEX and recovered by *Valeriana fauriei* [21]. Our results showed that a reduction of SOD activity by DEX was markedly recovered by RosA treatment (Figure 4C). These findings indicate that RosA is a good scavenger of ROS in DEX-treated C2C12 myotubes and may eventually inhibit muscle atrophy by DEX.

Apoptosis is generally associated with increased ROS production and causes the loss of muscle fiber nuclei, resulting in muscle atrophy [38,39]. Various types of muscle atrophy models induced by hindlimb suspension, sepsis, and aging have revealed that the atrophy of skeletal muscle tissues is associated with an increase in apoptosis [40,41,42]. BAX (B-cell lymphoma protein 2 (Bcl-2)-associated X) and Bcl-2 act as a promoter and an inhibitor of apoptosis, respectively. BAX is a critical downstream mediator of apoptosis, and it promotes cell death through the induction of the mitochondrial permeability transition [43]. In contrast, BAX activation is attenuated by Bcl-2 thereby inhibiting apoptosis. Several studies have shown that DEX-treated C2C12 myotubes or mice increased apoptosis compared to the control group, whereas substances such as myricanol, quercetin, fucoxanthin, and 20(s)-ginseonside-Rg3 reduced BAX expression and increased Bcl-2 expression, preventing DEX-induced apoptosis in the muscle atrophy models [20,22,23,44]. Similar to the previous studies, our results also show increased an expression of BAX and decreased expression of Bcl-2 by DEX treatment in C2C12 myotubes, and these changes in apoptosis-related markers were significantly reduced by RosA (1 and 5 μM) treatment (Figure 4C,D). We also confirmed the effect of RosA on DEX-induced apoptosis in C2C12 myotubes using an Annexin V-FITC and PI double staining. Annexin V/PI double staining is a commonly used method for the study of apoptotic cells [45]. It was reported that when C2C12 myotubes were treated with DEX, the number of Annexin V/PI positive cells increased compared to the control group [23]. In the present results, C2C12 myotubes treated with DEX significantly increased Annexin V/PI-positive myotubes compared to the control, but Annexin V/PI-positive myotubes increased by DEX were reduced by RosA (Figure 4F,G). These results demonstrate that RosA reduces apoptosis via a decrease of BAX and an increase of Bcl-2 in DEX-treated C2C12 myotubes and may eventually inhibit muscle atrophy by DEX.

Autophagy plays a critical role in removing altered organelles and degraded protein in skeletal muscle. The autophagy signaling pathway is essential for energy generation/consumption processes in skeletal muscles. The impairment of autophagy is associated with mitochondrial dysfunction, the accumulation of cell damage, and cell death, and it has been identified as one of the major causes of a variety of skeletal muscle disorders [46]. Shen et al. and Zhiyin et al. reported that DEX treatment induces atrophy by significantly inhibiting autophagy in C2C12 myotubes through a decrease in the LC3-II/LC3-I ratio and Beclin1 expression and an increase in p62 expression [20,23]. In the present study, the inhibition of autophagy by DEX in C2C12 myotubes was activated at both concentrations of RosA treatment (1 and 5 μM) (Figure 5A–D). Our results indicate that RosA protects DEX-induced muscle atrophy by restoring autophagy inhibition.

Mitochondrial content and ATP production activities are associated with muscle atrophy. Mitochondrial dysfunction promotes skeletal muscle wasting, and it appears in various models of muscle atrophy, including disuse, diabetes, and aging [32,47,48,49]. The normal range of mitochondrial content and ATP production capacity plays a critical role in maintaining mitochondrial function and muscle function [50,51]. Mitochondrial ROS is known to affect the induction of muscle atrophy by breaking the balance between protein degradation and synthesis, and skeletal muscle atrophy due to denervation and disuse is associated with increased mitochondrial ROS production [32,52,53]. Previous studies reported that DEX accelerates muscle atrophy by inducing mitochondrial dysfunction through decreased mitochondrial content and ATP production in C2C12 myotubes compared to the control [54]. In addition, Kim et al. reported that ginseng dried extract (BST 204) could prevent mitochondrial dysfunction and atrophy by reducing the formation of mtROS by DEX in C2C12 myotubes [26]. In this study, mitochondrial content and ATP levels were significantly decreased by DEX compared to controls in C2C12 myotubes (Figure 6A–C). However, these reductions were restored to a level similar to that of the control by the treatment of 5 µM RosA (Figure 6A–C). In addition, we found that RosA reduced DEX-induced mitochondrial ROS production in C2C12 myotubes (Figure 6D,E). These results indicate that RosA can protect against muscle atrophy by inhibiting mitochondrial dysfunction in DEX-treated C2C12 myotubes.

## 4. Materials and Methods

### 4.1. Plant Materials

The plant materials of *Salvia plebeia* R.Br were purchased from Kyungdong Oriental Market (Seoul, Republic of Korea) in October 2021 and identified by one of the corresponding authors (CY Kim). A voucher was deposited at the Pharmacognosy Laboratory of the College of Pharmacy, Hanyang University (specimen no. HYUP-SP-001). An amount of 279 g of dried *S. plebeia* was extracted three times with 4 L of 50% ethanol for 3 h at 60 °C, and the solvents were evaporated in vacuo at 40 °C, yielding the ethanol extract (14.9 g).

### 4.2. Reagents

Dulbecco’s Modified Eagle Medium (DMEM) 1640 medium, fetal bovine serum (FBS), horse serum (HS), and penicillin/streptomycin were purchased from Gibco Corporation (Morgan Hill, CA, USA). Dimethyl sulfoxide (DMSO), rosmarnic acid, dexamethasone, Hoechst 33342, Triton X-100, and Tween 20 were purchased from Sigma-Aldrich (St. Louis, MO, USA). A cell counting kit was purchased from Dojindo (Kumamoto, Japan). CM-H_2_DCFDA, propidium Iodide, Annexin-V, MitoTracker Deep Red, and MitoSOX Red were purchased from Invitrogen (Waltham, NY, USA). Phosphate-buffered saline (PBS) and 3% blocking buffers were purchased from Biosesang (Seongnam, Republic of Korea). Ethanol, *n*-hexane, ethyl acetate, and *n*-butanol were purchased from Daejung Chemical (Siheung, Republic of Korea).

### 4.3. C2C12 Cell Culture and Differentiation

C2C12 myoblast was purchased from the American Type Culture Collection (ATCC, Manassas, VA, USA) and cultivated in high-glucose (25 mM glucose) DMEM containing 10 % FBS and 1% penicillin-streptomycin in 5% CO_2_ at 37 °C. When a confluence of myoblasts of more than 90% was reached, C2C12 myoblasts were seeded onto 6-well plates (1.2 × 10^5^ cells/well), 12-well plates (6 × 10^4^ cells/well), and 48-well plates (1 × 10^4^ cells/well). To differentiate myoblasts into myotubes, when C2C12 myoblasts were 80–90% confluent, the growth medium was changed to differentiation medium (high-glucose DMEM containing 2% HS, 1% penicillin-streptomycin) for 6 days, and the medium was changed every 2 days.

### 4.4. Treatment of SPR, RosA, and Dexamethasone

After differentiation, myotubes were treated with 10 μM dexamethasone (DEX) in the presence or absence of different concentrations SPR (1, 5, and 10 μg/mL) or RosA (1 and 5 μM) for 24 h. The control group was incubated in DMEM supplement 2% HS and 0.1% DMSO, which is a DEX vehicle solution. After 24 h, the myotubes were washed with ice-cold PBS and then scraped and lysed with cell lysis buffer including a protease inhibitor cocktail and phosphatase inhibitor cocktail. Subsequently, all groups were harvested for the next experiments.

### 4.5. Measurement of Cell Viability

C2C12 myotube viability was assessed using a cell counting kit-8 (CCK-8) according to the manufacturer’s instructions. Briefly, C2C12 myoblasts (1 × 10^4^ cells/well) were seeded onto 48-well plates for 24 h and fully differentiated into C2C12 myotubes for 6 days. After differentiation, the C2C12 myotubes were treated with 10 μM DEX in the presence or absence of SPR (1, 5, and 10 μg/mL) or RosA (1 and 5 μM) for 24 h, respectively. Subsequently, 20 µL of CCK reagent was added to each well and followed incubation for 4 h at 37 °C. After incubation, the absorbance was measured using an EnSpire Multimode Plate Reader at 450 nm (PerkinElmer, Waltham, MA, USA).

### 4.6. Myosin Heavy Chain (MHC) Immunofluorescence Staining

C2C12 myotubes were fixed with 4% paraformaldehyde for 10 min at room temperature and permeabilized with 0.1% Triton X-100 for 20 min in phosphate-buffered saline (PBS). Subsequently, blocking was incubated with 3% bovine serum albumin (BSA) for 1 h at room temperature. After blocking, the myotubes were incubated with MHC primary antibody (1:300, Santa Cruz, SC-376157, Dallas, TX, USA) overnight at 4 °C. After washing 3 times with 0.1% phosphate-buffered saline-Tween 20 (PBST), the myotubes were incubated with a secondary antibody conjugated with Alexa Fluor 488 (1:500, Invitrogen, Waltham, MA, USA) at 37 °C for 1 h. After washing 3 times with 0.1% PBST, nuclei were counterstained with 10 µM of Hoechst 33342. The stained C2C12 myotubes were observed under a fluorescence microscope (JuLI^TM^ stage, Nano Entek, Seoul, Republic of Korea). The average of each myotube diameter was measured in a total of 100 myotubes from at least 5 different fields using ImageJ software (1.48 version). For the MHC-positive area analysis, 5 randomly selected fields were counted from three independent experiments in each group. The fusion index was calculated with the equation as follows: (number of nuclei inside MHC-positive myotubes in 5 fields)/(total number of nuclei present in 5 fields). The myotubes’ diameters, MHC-stained area, and fusion index were measured and analyzed using ImageJ software (1.48 version).

### 4.7. Western Blot Analysis

C2C12 myotubes were washed 3 times with ice-cold PBS and lysed with cold RIPA lysis buffer (Invitrogen, Waltham, MA, USA) containing 1% protease and 1% phosphatase inhibitor cocktails on ice for 30 min. Whole-cell lysates were then centrifuged at 12,000× *g* for 20 min, and supernatants were transferred into new tubes. After transfer, the protein concentration of each sample was determined using the Pierce BCA Protein Assay Kit (Thermo Fisher Scientific, St. Louis, MA, USA). Equal amounts of protein (20 µg) were loaded and separated on 8% or 12% SDS polyacrylamide gel electrophoresis and transferred through electroblotting to the PVDF membranes (Merck, Darmstadt, Land Hessen, Germany) for 1 h and blocked with 3% BSA solution for 2 h at room temperature. After blocking, the membranes were incubated overnight in different primary antibodies against MHC, FoxO3a, mTOR, MuRF1, Atrogin-1, BAX, Bcl-2, Beclin1, p62, and LC3B (1:1000, Santacruz, Dallas, TX, USA); ß-actin and GAPDH (1:5000, Santacruz, St. Louis, TX, USA); and Akt, p-Akt, p-mTOR, p70S6K, and p-p70S6K (1:1000, Cell Signaling Technology, Danvers, MA, USA) at 4 °C. Membranes were washed 3 times with 0.2% PBST and incubated with the appropriate horseradish peroxidase (HRP)-conjugated anti-mouse IgG (1:5000, Santacruz, Dallas, TX, USA) or anti-rabbit (1:5000, Santacruz, Dallas, TX, USA) IgG for 2 h at room temperature. Subsequently, the membranes were washed 3 times with 0.2% PBST. The protein bands of each membrane were visualized using enhanced chemiluminescence (Thermo Fisher Scientific, St. Louis, MA, USA) and Chemidoc imaging system (Biorad, Hercules, CA, USA), and densitometry analysis was performed using ImageJ software (1.48 version).

### 4.8. Measurement of Intracellular ROS

To evaluate intracellular ROS levels, we used a fluorescent probe, CM-H_2_DCFDA. The fully differentiated C2C12 myotubes were treated with 10 µM DEX in the presence or absence of RosA (1 and 5 µM, respectively) for 24 h. After treatment, the myotubes were washed with 3 times PBS, and 10 µM of CM-H_2_DCFDA solution was added to each well and incubated for 15 min in the dark at room temperature. The myotubes were washed 3 times with a culture medium and then photographed using a fluorescence microscope (JuLI^TM^ stage). CM-CFD-positive myotubes were counted in at least 5 randomly selected fields from each well in three independent experiments. The fluorescence intensity was analyzed using ImageJ software (1.48 version).

### 4.9. Measurement of SOD Activity

The SOD activities in C2C12 myotubes were analyzed according to the instructions of the SOD assay kit. Briefly, 20 µL of each sample was transferred into a 96-well plate, and 200 µL of WST working solution and 20 µL of enzyme working solution were added and incubated at 37 °C for 20 min. After incubation, SOD activity was measured at 450 nm using an EnSpire Multimode Plate Reader (PerkinElmer, Waltham, MA, USA).

### 4.10. Annexin V/PI Double Staining

C2C12 myoblasts were seeded onto 12-well plates (6 × 10^4^ cells/well). After differentiation, myotubes were treated with 10 μM DEX in the presence or absence of RosA for 24 h. After 24 h, the myotubes were washed twice with PBS and trypsinized. The myotubes were directly centrifuged and suspended in 1x binding buffer (500 μL). After suspension, Annexin V (5 μL), PI (5 μg/mL), and Hoechst (10 μg/mL) were added and incubated for 5 min in the dark at 37 °C. Stained myotubes were evaluated under a fluorescence microscope. Annexin-V/PI-positive myotubes were counted in at least 5 randomly selected fields from each well in three independent experiments. The fluorescence intensity was analyzed using ImageJ software (1.48 version).

### 4.11. Mitochondria Staining 

Fully differentiated C2C12 myotubes were treated with 10 µM DEX in the presence or absence of RosA (1 and 5 µM, respectively) for 24 h. Subsequently, myotubes were incubated with 500 nM Mito Tracker Deep Red for 30 min at 37 °C. After incubation, the cells were washed with PBS and fixed with 4% paraformaldehyde. Afterward, the myotubes were washed three times with PBS and then photographed using a fluorescence microscope (JuLI^TM^ stage). Red fluorescence-positive myotubes were counted in at least 5 randomly selected fields from each well in three independent experiments, and the fluorescence intensity was analyzed using ImageJ software (1.48 version).

### 4.12. Measurement of the ATP Level

ATP levels of C2C12 myotubes were determined using a luminescent ATP detection assay kit according to the manufacturer (Cayman Chemical, Ann Arbor, MI, USA). Briefly, the fully differentiated myotubes were treated with RosA (1 or 5 µM) and/or DEX (10 µM) for 24 h. After treatment, the cells were washed with cold phosphate-buffered saline, and the myotubes were lysed using ATP detection sample buffer. The lysates were centrifuged for 10 min at 13,000× *g*, and 10 µL of supernatant was transferred into each well of a 96-well white plate. Afterward, 100 µL of ATP reaction mix solution was added to each well and incubated at room temperature for 20 min. After the reaction, the ATP levels were measured using an EnSpire Multimode Plate Reader at 560 nm (PerkinElmer, Waltham, MA, USA).

### 4.13. Measurement of Mitochondrial Oxidation 

To evaluate mitochondrial ROS, the mitochondrial superoxide indicator MitoSOX Red was used according to the manufacturer’s instructions. The fully differentiated C2C12 myotubes were treated with 10 µM DEX in the presence or absence of RosA (1 and 5 µM, respectively) for 24 h. Then, the myotubes were washed twice with PBS and incubated with 5 µM MitoSOX Red solution for 30 min in the dark at 37 °C. The myotubes were washed 3 times with a culture medium and then photographed using a fluorescence microscope (JuLI^TM^ stage). MitoSOX Red-positive myotubes were counted in at least 5 randomly selected fields from each well in three independent experiments. The fluorescence intensity was analyzed using ImageJ software (1.48 version).

### 4.14. Statistical Analyses

All data are presented as the mean ± standard deviation (SD) for at least three independent experiments. Statistical significance was evaluated and determined by one-way analysis of variance (ANOVA) using GraphPad Prism 5.0 (GraphPad Software Inc., La Jolla, CA, USA), followed by Tukey’s post-hoc test. A *p*-value less than 0.05 was considered statistically significant.

## 5. Conclusions

In summary, our results demonstrate that SPR and RosA alleviate DEX-induced muscle atrophy in C2C12 myotubes by inhibiting protein degradation, ROS production, apoptosis, autophagy, and mitochondria dysfunction as well as improving SOD activity and muscle protein synthesis. These findings suggest that SPR and RosA may be potential therapeutic agents for the prevention of muscle atrophy. However, further studies investigating other possible anti-atrophy-related mechanisms and in vivo studies are required.

## Figures and Tables

**Figure 1 ijms-24-01876-f001:**
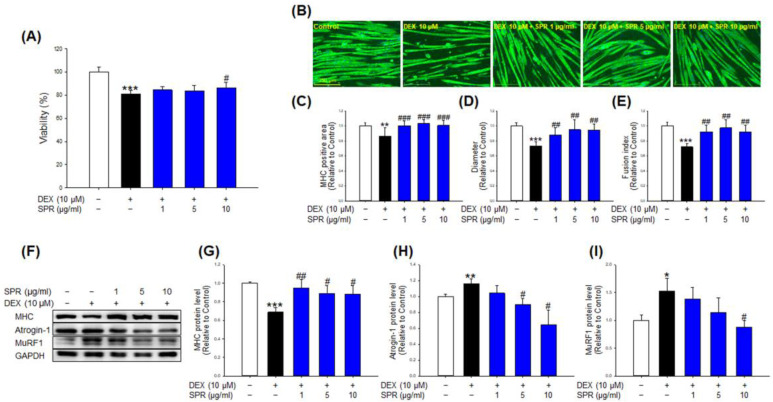
Effects of SPR on viability, morphology, and expression of muscle atrophy-related proteins in the DEX-induced C2C12 myotubes. (**A**) C2C12 myotubes were treated with DEX (10 μM) in the presence or absence of SPR (1, 5, and 10 μg/mL) for 24 h. (**B**) C2C12 myotubes were stained with MHC (green) and DAPI (blue), and representative photographs were observed under a fluorescent microscope (scale bar = 250 μm). (**C**) Relative change in MHC-stained area in myotubes, (**D**) diameters, and (**E**) fusion index were measured from randomly selected fields and were quantified using the Image J program. (**F**) Expressions of MHC, Atrogin-1, MuRF1, and GPADH were analyzed in C2C12 myotubes by western blotting. GAPDH was used as the loading control. (**G**) Quantitative analysis of MHC, (**H**) Atrogin-1, and (**I**) MuRF1. These results are presented as the means ± SD of three independent experiments: * *p* < 0.05, ** *p* < 0.01, *** *p* < 0.001 vs. control; ^#^
*p* < 0.05, ^##^
*p* < 0.01, ^###^
*p* < 0.001 vs. DEX.

**Figure 2 ijms-24-01876-f002:**
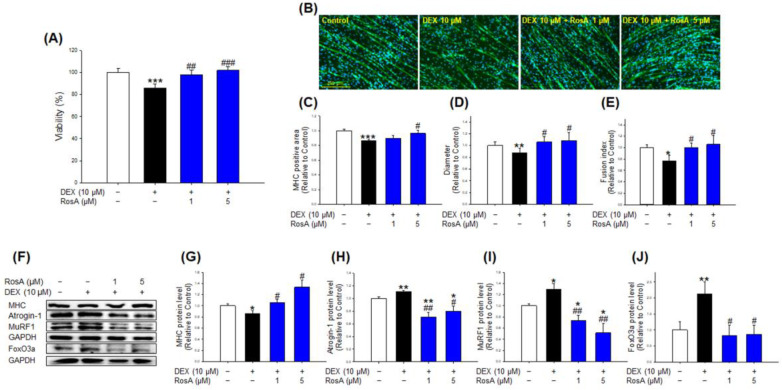
Effect of RosA on viability, morphology, and expression of muscle atrophy-related proteins in the DEX-stimulated C2C12 myotubes. (**A**) C2C12 myotubes were treated with DEX (10 μM) in the presence or absence of RosA (1 and 5 μM) for 24 h. (**B**) C2C12 myotubes were stained with MHC (green) and DAPI (blue), and representative photographs were observed under a fluorescent microscope (scale bar = 250 μm). (**C**) Relative change in the MHC-stained area in myotubes, (**D**) diameters, and (**E**) fusion index were measured from randomly selected fields and quantified using the Image J program. (**F**) Expressions of MHC, Atrogin-1, MuRF1, FoxO3a, and GPADH were analyzed in C2C12 myotubes by western blotting. GAPDH was used as the loading control. (**G**) Quantitative analysis of MHC, (**H**) Atrogin-1, (**I**) MuRF1, and (**J**) FoxO3a. These results are presented as the means ± SD of three independent experiments: * *p* < 0.05, ** *p* < 0.01, *** *p* < 0.001 vs. control; ^#^
*p* < 0.05, ^##^
*p* < 0.01, ^###^
*p* < 0.001 vs. DEX.

**Figure 3 ijms-24-01876-f003:**
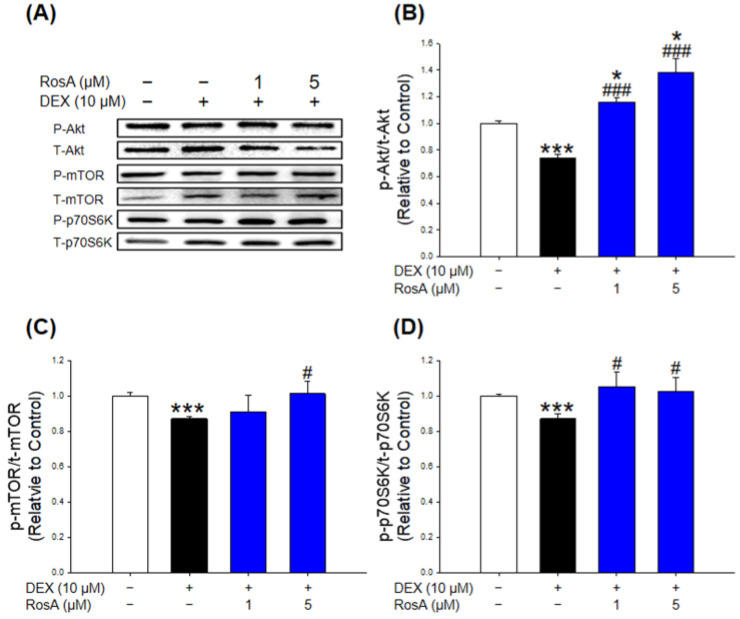
Effects of RosA on the Akt/mTOR/p70S6K pathway in DEX-mediated atrophy in C2C12 myotubes. (**A**) Western blot of p-AKT, AKT, p-mTOR, mTOR, p-p70S6K, p70S6K, and FoxO3a proteins in C2C12 myotubes treated with 10 μM DEX in the presence or absence of RosA (1 and 5 μM) for 24 h. (**B**) Quantitative analysis of the p-Akt/Akt. (**C**) Quantitative analysis of the p-mTOR/mTOR. (**D**) Quantitative analysis of the p-p70S6K/p70S6K. These results are presented as the means ± SD of three independent experiments: * *p* < 0.05, *** *p* < 0.001 vs. control; ^#^
*p* < 0.05, ^###^
*p* < 0.001 vs. DEX.

**Figure 4 ijms-24-01876-f004:**
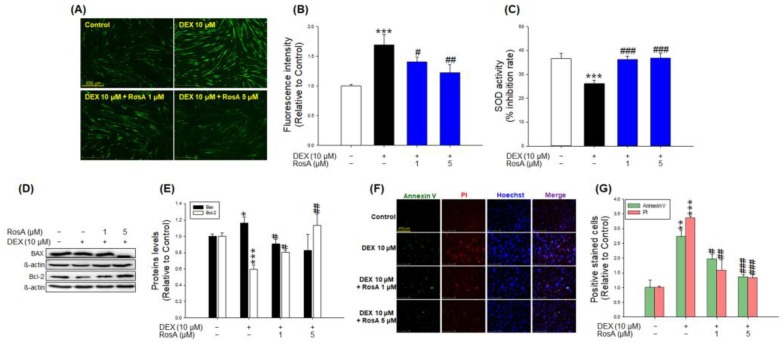
Effects of RosA on ROS production and expression of an apoptosis-related protein in atrophy-induced C2C12 by DEX treatment. (**A**) The production of cellular ROS analysis using a 2,7-dichlorodihydrofluorescein diacetate (DCFH-DA) dye and observed under a fluorescence microscope (scale bar = 500 μm). (**B**) The graph shows the quantification of the CM-H_2_DCFDA-stained area using the ImageJ software program (1.48 version). (**C**) The antioxidant enzyme (SOD) activity. (**D**) The expression of BAX and Bcl-2 was detected by western blotting analysis. (**E**) The quantitative of BAX and Bcl-2 expression levels. (**F**) Annexin V, PI, Hoechst staining of C2C2 myotubes (scale bar = 250 μm). (**G**) The graph presents Annexin V-positive and PI-positive myotubes. These results are presented as the means ± SD of three independent experiments: * *p* < 0.05, ** *p* < 0.01, *** *p* < 0.001 vs. control; ^#^
*p* < 0.05, ^##^
*p* < 0.01, ^###^
*p* < 0.001 vs. DEX.

**Figure 5 ijms-24-01876-f005:**
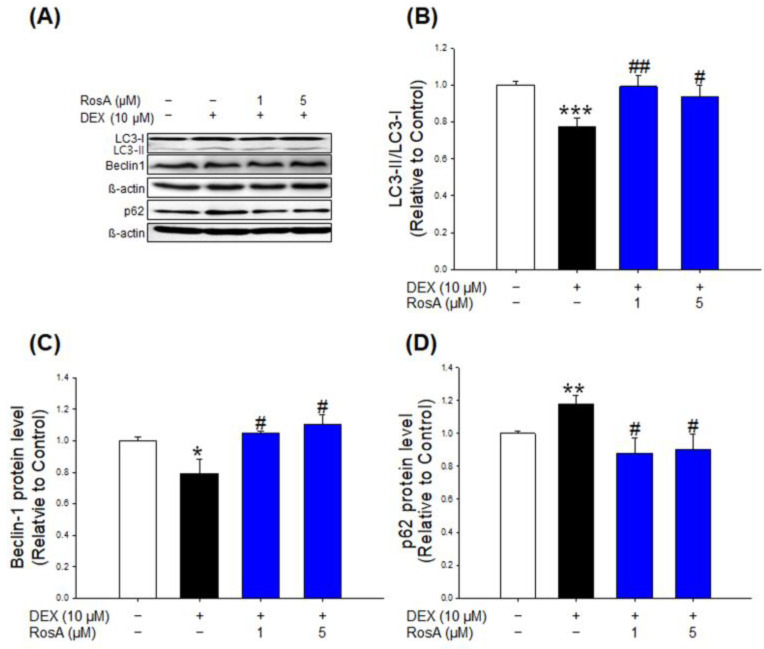
Effects of RosA on autophagy in DEX-treated C2C12 myotubes. (**A**) The expression of autophagy-associated proteins such as LC3, Beclin1, and p62. GAPDH was used as a loading control. (**B**) The quantitative of LC3II/LC3I ratio, (**C**) Beclin1, and (**D**) p62 expression levels. The graph shows a quantitative representation of the levels of protein. These results are presented as the means ± SD of three independent experiments: * *p* < 0.05, ** *p* < 0.01, *** *p* < 0.001 vs. control; ^#^
*p* < 0.05, ^##^
*p* < 0.01 vs. DEX.

**Figure 6 ijms-24-01876-f006:**
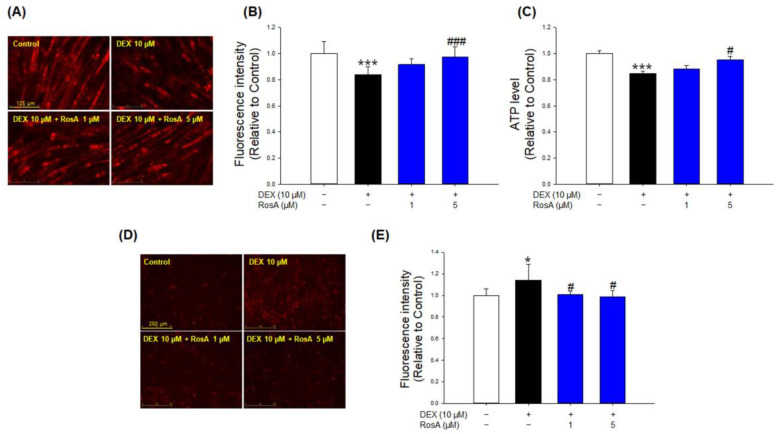
Effects of RosA on the mitochondrial content and ROS and ATP levels in DEX-treated C2C12 myotubes. (**A**) The mitochondrial contents were determined by MitoTracker Deep Red staining and observed by a fluorescence microscope (scale bar = 125 μm). (**B**) The graph shows the quantification of the MitoTracker Deep Red stained area by the ImageJ software program (1.48 version). (**C**) ATP production in the C2C12 myotubes. (**D**) Fluorescence images from MitoSOX Red in C2C12 myotubes (scale bar = 250 μm). (**E**) Quantification of the MitoSOX Red fluorescence-stained area. These results are presented as means ± SD of three independent experiments: * *p* < 0.05, *** *p* < 0.001 vs. control; ^#^
*p* < 0.05, ^###^
*p* < 0.001 vs. DEX.

## Data Availability

Not applicable.

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
