# Peer review of "Salvia plebeia R.Br. and Rosmarinic Acid Attenuate Dexamethasone-Induced Muscle Atrophy in C2C12 Myotubes"

_ijms, 2023, doi:10.3390/ijms24031876_

Round 1
Reviewer 1 Report
Introduction
The state of the art in the area studied must be improved, explaining actual knowledge and the necessity of the present research
Objective must be concise, according to the introduction proposed
Include a study hypothesis
Methods
Include methods section after introduction
For better comprehension procedure must be in deep described and I would recommend including a figure to visually explain procedure
Results
Well described and redacted
Include methods section after methods
Discussion
Include methods section after results
Explain if hypothesis is confirmed and why
Discuss results obtained and explain differences according to previous studies
Include practical applications and limitation of the study
Conclusion
Must be concise, responding the study aims
Reviewer 2 Report
Kim et al. presented results showing that DEX-induced muscle atrophy could be minimalized in the presence of rosmarinic acid (RA). The authors introduced several analyses demonstrating the protective effects of RA on C2C12 cells viability, expression of atrophy-associated proteins, atrophy-associated proteins, ATP synthesis, apoptosis, and autophagy. Concerning apoptosis and autophagy analysis, I have specific comments.
- There are several assays allowing apoptosis assessment. The analysis of BCl-2 family proteins is not the direct apoptotic assay. Changes in the expression of Bcl-2 family proteins could be related to the apoptosis process but do not indicate apoptosis itself. The authors should assess the apoptosis induction in control and RA-treated cells using effector caspases activity assays, or Annexin V/PI staining or other available.
- Autophagy activation should be assessed in accordance to accepted guidelines. Thus, the autophagy activation should be examined as an autophagic flux, along with the presence of, for example, chloroquine or another inhibitor. Then, the MAP-LC3 processing should be analyzed.
Minor comments:
- For all reagents, company name, city, state, and country should be provided. Please, update the Materials and Methods section.
Reviewer 3 Report
In the manuscript entitled ‘Salvia plebeia R.Br. and rosmarinic acid attenuate dexamethasone-induced muscle atrophy in C2C12 myotubes”, the authors studied thevarious signalling pathways involved in the protective effect of SPR and more particularly rosmarinic acid against muscle atrophy.
The authors tested the effect of SPR and rosmarinic acid (RosA) on dexamethasone (DEX)-induced skeletal muscle atrophy in C2C12 myotubes. Nevertheless, the authors did not present the effect of SPR and rosmarinic acid in C2C12 myotubes without DEX treatment. Because SPR and rosmarinic acid might have a hypertrophic effect on basal c2c12 myotubes, the authors need to add these control experiments in supplemental figures.
Furthermore, the authors presented a large amount of data but the link between the different results need to be strengthened.
Part 2.2: Line 132 The authors wrote “the expression level of atrogin-1 and MuRF1 protein was lower than that of the control group”. Were these differences significant? If yes, stars should be added figure 2H and I.
Part 2.4 and 2.6: the authors investigated the antioxidant effect of RosA and used only the general probe DCFH-DA. As mitochondria is one of the main source of oxidant in muscle cells, the authors should complete the study by testing mitosox that stained superoxide produced by mitochondria. Part 2.4 and 2.6 could be put together. Furthermore, the authors should also complete their study by evaluating the levels of antioxidant enzymes such as Cu/Zn SOD, Mn SOD and catalase.
On the other hand, the authors observed a positive effect of RosA on mitochondrial content. Il will be interesting to study the effect of RosA on mitochondrial biogenesis and evaluate peroxisome proliferator-activated receptor coactivator 1-α (PGC-1α), an upstream regulator of mitochondrial biogenesis and SIRT-1.
Part 2.5: the author studied the effect of RosA on autophagy in DEX-treated myotubes. They evaluated the levels of expression of LC3II/LC3 I, Beclin, p62. They should also studied markers of mitophagy such as Pink1 and Parkin that are known to be modulated by Rosmarinic acid (Nan Fang Yi Ke Da Xue Xue Bao . 2020 Nov 30;40(11):1628-1633. doi: 10.12122/j.issn.1673-4254.2020.11.14.)
Figures: to facilitate the reading, the size of all the figure labels should be increased.
In Figure 1,2,3,4 and 5, Standard deviation of the control group should be added.
Furthermore, scale bars should appear on each image of each figure.
Figure 1B: dapi staining should be added like in figure 2B.
Figure legends: atrogin-1 should be written with only one r (line 105, line 144).
Figure 3B: PAKT/AKT seems to be higher in RosA 1 and 5 treated myotubes than in control. If the difference is significant, the authors should add stars on the figure.
Figure 5 Line 206: LC3II/LCI ratio should be replaced by LC3II/LC3I ratio
Line 336: the sentence should be corrected
Materials and methods:
Part 4.3: the authors wrote “the growth medium was changed to differentiation medium (high-glucose DMEM containing 2% HS, 1% penicillin-streptomycin) for 6 days”. When did the authors performed DEX treatment? after 6 days of differentiation? Were the DEX and SPR or RosA treatment performed at the same time? And why?
Part 4.4: The authors used HS as abbreviation. Could they define it?
Part 4.6: MHC protein has various isoforms. Did the author use a general MHC antibody or which isoform is targeted? The authors should add the MHC antibody reference number in the text.
Line 395: The authors should use the word “myotubes” instead of “tubes”.
The authors should explain how they measured myotubes diameters.
Part 4.8 and 4.9: the authors wrote “positive cells were counted in at least 5 randomly selected fields from each well in three independent experiments. The fluorescence intensity was analysed using Image J software”. Could the authors mention the number of myotubes studied per experiments? They should also replace the word “cells “ (that could be used for myoblasts and myotubes) by myotubes (line 427, 429, 436, 438).
Line 429: Intracellular reactive oxygen species (ROS) staining was performed with chloromethyl 2′,7′-dichlorodihydrofluorescein diacetate (CM-H2DCFDA; Invitrogen). CM-H2DCFDA is hydrolyzed by nonspecific esterases to release 2′,7′-dichlorodihydrofluorescein (CM-H2DCF), which is oxidized by intracellular ROS, such as hydrogen peroxide, to CM-DCF, which emits green fluorescence. Therefore, CM-H2DCFDA positive cells should be replaced by CM-DCF positive myotubes.
Conclusion
Line 461: The sentence “Especially, RosA, a major constituent of SPR, inhibits protein degradation, ROS production, apoptosis, autophagy, and mitochondria dysfunction as well as improves muscle protein synthesis” should be replaced by “Especially, RosA, a major constituent of SPR, inhibits protein degradation, ROS production, apoptosis, autophagy, and mitochondria dysfunction as well as improves muscle protein synthesis induced by dexamethasone”.
Round 2
Reviewer 2 Report
I appreciate the Authors' responses and updates. I still have a major comment regarding apoptosis. PI staining is, in general, dedicated to necrotic cells detection. To discriminate apoptosis and necrosis, Annexin V/PI staining should be applied.
Author Response
Dear reviewers,
Thank you very much for your consideration, and we really appreciate the comments and have learned a lot. Appropriate changes were made and marked Red in the revised manuscript according to the suggestions of reviewers.
Response: Thanks for the reviewer’s good comments and suggestions, and we fully agree with the reviewer’s comments. According to the reviewer’s comments, we performed an Annexin V/PI staining. As shown in Figures 4F and G, Annexin V and PI-positive stained cells were significantly increased by DEX treatment, but RosA ameliorated DEX-induced apoptosis in C2C12 myotubes. Reagent information, method, result, figure, figure legend, discussion, and reference related to Annexin V/PI staining are written in each section of the manuscript.
Reviewer 3 Report
The authors revised the manuscript in accordance with comments.
Author Response
Dear reviewers,
Thank you very much for your consideration, and we really appreciate the comments and have learned a lot. Appropriate changes were made and marked blue in the revised manuscript according to the suggestions of reviewers and the editor.
Response: Thank you for your recognition.
Round 3
Reviewer 2 Report
The Authors updated the manuscript according to the previous comments. Apoptosis assay has been introduced thus, the presented results now support the conclusions. In my opinion, it significantly strengthened the value of the presented data.